# Effect of Solvent on Fluorescence Emission from Polyethylene Glycol-Coated Graphene Quantum Dots under Blue Light Illumination

**DOI:** 10.3390/nano11061383

**Published:** 2021-05-24

**Authors:** Po-Chih Yang, Yu-Xuan Ting, Siyong Gu, Yasser Ashraf Gandomi, Jianlin Li, Chien-Te Hsieh

**Affiliations:** 1Department of Chemical Engineering and Materials Science, Yuan Ze University, Taoyuan 32003, Taiwan; danny0935331003@gmail.com; 2Fujian Provincial Key Laboratory of Functional Materials and Applications, School of Materials Science and Engineering, Xiamen University of Technology, Xiamen 361024, China; gu-siyong@163.com; 3Department of Chemical Engineering, Massachusetts Institute of Technology, Cambridge, MA 02142, USA; ygandomi@mit.edu; 4Oak Ridge National Laboratory, Electrification and Energy Infrastructure Division, Oak Ridge, TN 37831, USA; lij4@ornl.gov

**Keywords:** nitrogen functionalization, graphene quantum dots, photoluminescence, hydrothermal synthesis

## Abstract

To explore aggregate-induced emission (AIE) properties, this study adopts a one-pot hydrothermal route for synthesizing polyethylene glycol (PEG)-coated graphene quantum dot (GQD) clusters, enabling the emission of highly intense photoluminescence under blue light illumination. The hydrothermal synthesis was performed at 300 °C using o-phenylenediamine as the nitrogen and carbon sources in the presence of PEG. Three different solvents, propylene glycol methyl ether acetate (PGMEA), ethanol, and water, were used for dispersing the PEG-coated GQDs, where extremely high fluorescent emission was achieved at 530–550 nm. It was shown that the quantum yield (QY) of PEG-coated GQD suspensions is strongly dependent on the solvent type. The pristine GQD suspension tends to be quenched (i.e., QY: ~1%) when dispersed in PGMEA (aggregation-caused quenching). However, coating GQD nanoparticles with polyethylene glycol results in substantial enhancement of the quantum yield. When investigating the photoluminescence emission from PEG-coated GQD clusters, the surface tension of the solvents was within the range of from 26.9 to 46.0 mN/m. This critical index can be tuned for assessing the transition point needed to activate the AIE mechanism which ultimately boosts the fluorescence intensity. The one-pot hydrothermal route established in this study can be adopted to engineer PEG-coated GQD clusters with solid-state PL emission capabilities, which are needed for next-generation optical, bio-sensing, and energy storage/conversion devices.

## 1. Introduction

Graphene quantum dots (GQDs) have experienced a surge in attention compared to other nanomaterials thanks to their superior physicochemical properties including excellent solubility, high stability, and low toxicity [1]. In particular, due to their stable photoluminescence and ultra-high resistance to photobleaching, GQDs have been widely used in the architecture of various devices such as bio-imaging instruments [2], high-precision sensors [3], electrochemical energy devices [4,5,6,7,8,9,10], light-emitting apparatus [11], and solar reactors [12]. It is well established that GQDs with an average particle size of <10 nm exhibit quantum confinement, and as a result, photoluminescence (PL) with variable wavelength and photobleaching impedance can be designed and engineered [13]. Compared to conventional semiconducting quantum dots, GQDs have superior properties such as remarkable biocompatibility, very robust interfacial structure, high solubility (aqueous), and a tunable band gap [11,14].

Despite these remarkable attributes, the widespread adoption of GQDs faces several critical challenges. Low quantum yield (QY), wide full width at half maximum (FWHM), and solvent-dependent PL behavior are among the major issues yet to be addressed. The latter plays a crucial role in affecting the PL behavior of GQD suspensions such as red/blue shift, FWHM, and QY, mainly originating from its dispersion extent and storage period in various solvents. One important PL mechanism, aggregation-caused quenching (ACQ), has recently been discovered, where the molecules are aggregated due to the intense intermolecular π–π interactions [15,16]. The conventional organic dyes commonly demonstrate ACQ behavior; thus, their implementation within the organic light emitting diodes as well as the bio-imaging devices has been significantly limited since the emission is easily quenched when an organic dye is employed as a solid film and in the aggregate state [15]. On the other hand, aggregate-induced emission (AIE) is intense for the solid state, as well as the aggregated mode, and is insignificant in dilute solutions. Most commonly, restricting the intramolecular motions (RIM) is considered a key control for modulating the AIE operation [17]. To further explore the RIM structures, several prior efforts have focused on developing hydrothermal synthesis routes for preparing polyethylene glycol (PEG)-passivated carbon nanostructures in the form of carbon nanodots [18]. It has been shown that the QY associated with the PEG-passivated composite nanostructures is almost two-fold higher compared to the pristine case with no passivation [18]. Recently, our group adopted a one-pot solvothermal method for synthesizing PEG-coated GQD clusters, capable of emitting green (QY: 85.3%) and red (14.6%) fluorescence under blue light illumination [19]. Such an ultrahigh QY for the PEG-coated GQD suspensions was primarily due to the formation of a structurally robust, two-dimensional framework on the exterior surface of the GQDs (i.e., a PEG-coated aggregate).

In our previous work, we demonstrated the capability of the PEG-coated GQD clusters of emitting green and red fluorescence under blue light illumination; however, an in-depth analysis of the AIE-enhanced luminescent process in PEG-coated GQD nanostructures is yet to be conducted. Therefore, this study aims to explore the influence of solvent on the PL performance of PEG-coated GQD clusters. Herein, we report a facile and green hydrothermal technique to produce PEG-coated GQD clusters using o-phenylenediamine (o-PD), as both the nitrogen and carbon sources in the presence of PEG. Pioneering studies have proved the efficacy of o-PD precursors for synthesizing carbon nanodots using hydrothermal [20,21], solvothermal [22], microwave-assisted [23], and electrochemical bottom-up synthesis techniques [24]. The PEG chains enable the in-situ formation of a two-dimensional network covering the GQDs during the hydrothermal molecular fusion process. The effect of the molecular weight of PEG on the PL emission from the PEG-coated GQD clusters was explored in water, ethanol, and propylene glycol methyl ether acetate (PGMEA). Finally, we have proposed a single-step hydrothermal synthesis route that can be employed for preparing immensely luminescent PEG-coated GQD clusters. Thus, this work has been dedicated to establishing the ACQ/AIE cycle through exploring the relationship between the solvent type and cluster structure, based on the RIM fluorescence mechanism.

## 2. Materials and Methods

### 2.1. Hydrothermal Synthesis of PEG-Coated GQD Clusters

To synthesize the PEG-coated GQD clusters, first, the *o*-PD (1.5 g, Alfa Aesar, Ward Hill, MA, USA, purity: 98%,) and PEG (1.5 g, purity: 99.5% Sigma Aldrich, St. Louis, MO, USA) were homogenously dispersed in 50 mL deionized water, then thoroughly stirred to prepare a well-mixed solution. Second, the suspension was placed into a poly(tetrafluoroethylene) (Teflon)-lined autoclave (75 mL, Macro Fortunate Co., Ltd., Taipei, Taiwan). The hydrothermal synthesis procedure was conducted at 300 °C for ~120 min. Upon cooling down to the ambient temperature, the as-prepared suspension with PEG-coated GQD samples was freeze-dried at −30 °C for 72 h. The PEG-coated GQD samples were filtered using a microporous separator with the average pore size of ~0.02 μm in order to remove any insoluble residuals. Finally, the as-purified CNDs were dispersed in ultrapure water before conducting any experiment. Schematic illustration of synthesis procedure for the pristine and PEG-coated GQD clusters is provided in Figure 1. Throughout this paper, the pristine GQD samples with no PEG polymer additives have been labeled as “GQD-P”. In addition, the “GQD-100k”, and “GQD-6k” symbols have been used referring to the as-prepared PEG powders with higher molecular weight (i.e., 100,000 and 6000, respectively).

### 2.2. Materials Characterization of PEG-Coated GQD Clusters

To characterize the as-prepared PEG-coated GQD clusters, several characterization methods and apparatus were employed, including a transmission electron microscopy (HR-TEM, model: F200s/Talos/FEI, Middlesex County, MA, USA), X-ray photoelectron spectroscopy (XPS, model: ESCA210/Fiscon VG, AZ, USA), Fourier-transform infrared (FT-IR, Varian, 1000-FTIR, Palo Alto, CA, USA), and ultraviolet-visible (UV-vis, Jasco, v650, Tokyo, Japan) absorption spectroscopy. To conduct the HR-TEM spectroscopy, the FEI Talos F200s electron microscope was used at 200 kV. The XPS spectra were recorded with the Fison VG ESCA210 machine and the corresponding C1s, N1s, and O1s peaks were fitted using an optimization algorithm developed in-house. 

In order to prepare the GQD suspensions, the GQD powder (50 mg) was uniformly dispersed in different solvents (500 mL) including deionized water, PGMEA (Sigma Aldrich, St. Louis, MO, USA, purity: 99.5%), and ethanol (ECHO Chem., Miaoli, Taiwan, purity: 85 vol%). Next, a fluorescence spectrometer (Hitachi F-7000 FLS920P, Taipei, Taiwan) was utilized (at 450 and 360 nm) for acquiring the PL emission spectra. Finally, the quantum yield for each solution was assessed with respect to the Coumarin (C_9_H_6_O_2_) reference (i.e., quantum yield ~73% at 373 nm) using Equation (1) [11]
QY = QY_r_ × [(PL_a_/OD)_s_/(PL_a_/OD)_r_] × Փ_s_^2^/Փ_r_^2^(1)
where the subscripts “r” and “s” are used for the reference and the GQD sample, respectively. Furthermore, Փ represents the reflective index of the solvent, “PL_a_” refers to the PL emission spectral surface, and “OD” is the absorbance magnitude.

## 3. Results and Discussion

HR-TEM micrographs of the GQD samples are provided in Figure 2. The GQD-P sample, as shown in Figure 2a, displays a spherical shape with an average diameter of ~3 nm. According to Figure 2a, a well-resolved lattice spacing of 0.21 nm, corresponding to the (100) facet of graphite [19,25], can be identified, indicating the formation of polycrystalline or amorphous carbon nanodots. The selected area diffraction (SAD) pattern (see the inset of Figure 2a reveals that the single GQD-P powder is of polycrystalline domain due to the presence of circular rings in the SAD pattern [19]. Figure 2b,c includes the HR-TEM micrographs of PEG-coated GQD samples, confirming the presence of GQD clusters induced by the PEG bonding. It is critical to mention that the GQD nanoparticles have a high tendency to be confined together; however, being well-dispersed within the solution enables the formation of the PEG-coated clusters. The GQD nanoparticles formed demonstrate a relatively uniform particle size distribution where the diameter of the nanoparticles was in the range of 3–5 nm. Additionally, the cluster size increases with increased PEG molecular weight, i.e., 310 nm (PEG-100k) > 28 nm (PEG-6k). The SAD patterns, as illustrated in the inset of Figure 2a–c, also confirm the hydrothermal synthesis of polycrystalline structure of the GQD nanoparticles within the clusters. To further explore this observation, a dynamic laser-scattering (DLS) technique was used to characterize the cluster size of the PEG-coated composites, as depicted in the electronic supporting information (see Appendix A). According to Appendix A, the cluster size dependence on the PEG molecular weight can be identified (i.e., a trend of increasing cluster size with increasing molecular weight). This result confirms that the in-situ hydrothermal process provides a straightforward pathway for engineering the spherical carbon nanodots along with the PEG-coated clusters during the one-pot synthesis process [19].

The XPS measurement was carried out to explore the surface chemistry and the distribution of functionalities imparted by the PEG coating layer. As shown in the survey-scan XPS spectra (see Appendix A), all the GQD samples were comprised of three main elements including C1s (ca. 282–292 eV), N1s (ca. 396–405 eV), and O1s (ca. 530–535 eV) [26,27]. The pristine GQD sample contained oxidation and amidation levels of 53.6 (O/C ratio) and 20.2 at.% (N/C ratio), respectively. Upon being modified by PEG bonding, the oxidation extent of PEG-coated GQD clusters decreased to 48.4 (GQD-6k) and 39.1 at.% (GQD-100k), while the amidation extent drastically reduced to 4.6 (GQD-6k) and 3.3 at.% (GQD-100k), as depicted in Appendix A. The changes in oxidation/amidation levels were mainly due to the formation of the PEG skin layer that covers the GQD particles and constructs a stereo framework of H−(O−CH_2_−CH_2_)_n_−OH chains. The decreased O/C ratio after the PEG coating is presumably due to polymer chains containing a large number of alkyl groups that tend to totally cover O-rich GQD surfaces (i.e., GQD-P (O/C ratio: 53.6 at.%)). We also observed that the O/C ratio is a decreasing function of PEG content. Thus, the construction of such a polymeric framework results in a lower amidation level within the molecular structure. It is worth mentioning that the GQD-100k sample imparts the lowest N/C atomic ratio among the GQD samples, attributed to the fact that high-molecular-weight PEG acts as a protective layer covering the GQD clusters, as confirmed by the HR-TEM and DLS analyses.

The distribution of oxygen functionalities on the pristine and PEG-coated GQD nanoparticles was investigated through decomposing the C1s and N1s peaks with an algorithm developed in-house (see Figure 3). First, the C1s spectra were extracted into five peaks at C=C/C‒C (ca. 284.5 eV), C‒N (ca. 285.8 eV), C‒O (ca. 286.2 eV), and C=O (ca. 287.2 eV) [27,28]. It is very clear that the oxygen functionalities decrease upon the introduction of PEG chains into the GQD clusters. Indeed, the PEG framework provides a large amount of H−(O−CH_2_−CH_2_)_n_−OH chains, covering the GQD solid state. Due to the PEG skin layer, the C‒O/C=O ratio shows an increasing trend with the PEG content, i.e., GQD-P (0.59) < GQD-6k (1.63) < GQD-100k (1.86). Figure 4 illustrates the N1s peaks deconvoluted into three major peaks located at 399.6 eV (pyrrolic or pyridinic N), 400.4 eV (quaternary N), and 401.5 eV (N-oxides) [29,30]. The first and second components are due to the presence of aromatic C=N–C and tertiary N‒(C)_3_ bonds, respectively [31,32]. The presence of these two N doping types, commonly called “lattice-N”, confirms that the N-doping onto the GQD lattices was successful. As shown in Figure 5, the O1s spectra are composed of two major peaks at ca. 531.8 and ca. 532.5 eV, which mainly originated from the presence of C=O and C–O bonds, respectively [33].

FT-IR spectra were also recorded to examine the functional groups formed on the surface of the GQD samples (see Appendix A). We observed that one transmittance peak in the range of 1200–1300 cm^−1^ corresponded to the C‒O stretch of –COOH, revealing the appearance of hydroxyl and carboxyl functional groups [34,35,36] attached to the GQD clusters. The transmittance band appearing at ~1710 cm^−1^ is primarily due to the presence of C=N or C=O surface functionalities [37,38]. Furthermore, a single band at ca. 3250–3400 cm^−1^ was observed due to the adsorption of water molecules or N‒H groups on the surface of GQD clusters [39]. Due to the hydrophilic PEG skin layer, the transmittance band becomes more evident on the PEG-coated GQD clusters compared to the pristine GQD sample, indicating the presence of a water-adsorbed layer onto the PEG-coated GQD clusters. The analysis of FT-IR spectra is consistent with the insight gained through the XPS measurements.

Figure 6a–c includes the UV-vis absorbance spectra of various GQD suspensions in DI water, PGMEA, and ethanol. The UV-vis spectra contain three major absorption bands with wavelengths ranging from 200 to 500 nm. The absorption bands at 200–250 nm are usually due to the π→π* transition of C=C of the sp^2^ domain and the absorption bands within 280–450 nm can be attributed to the n→π* transition of C=N and C=O bonds [39,40,41]. Compared to the GQD-P suspension, both PEG-coated GQD clusters (i.e., GQD-6k and GQD-100k) exhibit an obvious shift in the C=C absorption band (π→π*) and an intense absorption band associated with the n→π* transition. This confirms that the C-containing bonds in the molecular structure (C=N/C=O, C–O, and C−N) promote the n→π* transitions within solid or aggregate states [42]. Indeed, the PEG layer covers the GQD core, leading to a substantial absorbance throughout the entire domain. It is important to mention that the absorbance edge (n→π*) in PGMEA demonstrates a different absorbance pattern compared to that in water and ethanol. Indeed, it can be inferred that the polarity of the solvent influences the absorbance of the GQD clusters [11,43]. This trend is mainly due to partial dissolution of hydrophilic PEG nanolayer in high-polarity solvents (i.e., water), whereas the PEG layer constructs a protective layer against PGMEA molecules, creating a robust framework of PEG-coated GQD clusters. Therefore, the formation of the PEG layer is influenced by the solvent type and is capable of altering the edge morphology at multiple band gap structures.

Figure 7 depicts PL emission spectra of the as-prepared and PEG-coated GQD clusters recorded within the range of 360–450 nm in water, ethanol, and PGMEA. As shown in Figure 7, the GQD suspensions in water (see Figure 7a,d) display a quasi-symmetric peak in the range of 650–700 nm. The PL spectrum includes a single band at ca. 570 nm with a FWHM of ca. 90 nm under 360 and 450 nm. Compared to the emission spectra associated with the GQD suspensions in ethanol (see Figure 6b,e), the GQD-P suspension exhibits an asymmetric lump at 450 nm, while both GQD-6k and GQD-100k samples demonstrate a quasi-symmetric peak with a small FWHM of 95 nm. Since the PEG skin layer covers over the GQD samples, the maximal PL emission shows a slight red-shift from 540 to 550 nm. For the GQD suspensions in PGMEA (as shown in Figure 7c,f), the GQD-P sample displays a very weak fluorescence, being quenched at 360 and 450 nm. In contrast, the GQD-6k and GQD-100k suspensions still emit ~530-nm light under 360 and 450 nm. Accordingly, the solvent type significantly alters the PL emission (e.g., fluorescence intensity) from the PEG-coated GQD clusters, while the influence of solvent type on the maximal wavelength (i.e., blue/red shift) is insignificant. This finding also demonstrates that the PEG-coated GQD clusters are capable of emitting intense fluorescence under blue light illumination (i.e., 450 nm).

To further explore the effect of solvent type on the PL emissions, QY was assessed as a crucial index in evaluating the PL performance from the pristine GQD and PEG-coated GQD clusters. Figure 8a–c includes the QY values (see Equation (1)) as a function of different GQD structures under UV light illumination. Analyzing the GQD suspensions in water and ethanol, the QY values ranged from 15.2% to 21.6%. Notably, the GQD-P suspension in PGMEA was substantially quenched with an ultralow quantum yield (QY < 2.5%). However, the magnitude of QY in PGMEA was significantly enhanced after the introduction of PEG coating. A similar trend was observed when the GQD suspensions were placed under blue light illumination (i.e., 450 nm), as illustrated in Figure 8d–f. According to Figure 8, the QY values were ultra-high (60–95%) for the GQD suspensions in water and ethanol. Nonetheless, the QY values drastically decreased when the GQD-P suspension was placed in PGMEA. Implementing the PEG coating, the QY values were significantly enhanced for the GQD suspensions in PGMEA.

This enhanced QY associated with the PEG-coated GQD clusters is mainly due to the formation of robust PEG-coated GQD aggregates that severely restrict the intermolecular rotational motion and subsequently boost the PL intensity [44]. It is important to note that the PGMEA is frequently used in the structure of various optical devices (e.g., micro-light-emitting diode display). Therefore, the PEG-coated GQD clusters can potentially enable solid-state fluorescence feasibility for next-generation optical instruments if the recipe for RIE clusters is optimized.

The PL emission from the GQD suspensions demonstrates a significant effect of the solvent type on the PL performance. Optical photographs of all GQD suspensions in different solvents under blue-light illumination (450 nm) were also collected, as shown in Appendix A. This implies that the selection of the solvent type not only alters the PL behavior but also affects the AIE fluorescence through the RIE mechanism. Considering the solvents used in this work (deionized water (surface tension: ~72.3 mN/m, pH = 5.8), ethanol (surface tension: ~46.0 mN/m, pH = 7.1), and PGMEA (surface tension: ~26.9 mN/m, pH = 7.5)), a large number of oxygen and nitrogen functional groups are formed on the edge of GQD-P nanoparticles serving as the electron donor groups [45,46]. Indeed, the formation of hydrophilic edges on the functionalized GQDs facilitates the adsorption of polar molecules onto the surface by providing emissive traps within the molecular structure [47]. This analysis is consistent with the PL emission mechanism where the GQD-P suspension in high-polar solvents, i.e., water and ethanol, displays a highly luminescent green emission with increased QY values. In contrast, the remarkable PL emission recorded for the PEG-coated GQD clusters in PGMEA is primarily due to the decreased solvent polarity [43] (i.e., polarity index: H_2_O >> PGMEA) along with reduced solubility, resulting in the rigid confinement of GQDs within the PEG framework. This solvent-dependent mechanism is similar to the “solvatochromism” behavior that organic dyes commonly exhibit [48,49,50]. Since the surface tension of the solvent ranges from 26.9 to 46.0 mN/m, the transition point can be reached for activating the AIE mechanism, which ultimately results in an improved fluorescence intensity.

A schematic diagram illustrating the PL emission from the GQD-P and GQD-100k suspensions in water and PGMEA is illustrated in Figure 9. Decorating the O atoms, along with doping the N atoms within the GQD-P sample, enables the formation of a superhydrophilic carbon surface and provides strong affinity to the water molecules. Considering the illumination at 450 nm, upon electron transition, an electron–hole pair is generated when the photons are absorbed by the double bonds within the molecular structure [19]. The transition of the excited electrons from a higher energy level (i.e., orbital) to a lower state results in substantial PL emission where various excited electronic states are observed (i.e., N_π_*, O_π_* and C_π_*) [19,26]. As shown in Figure 8a, the GQD-P suspension in water enables a strong fluorescence due to being well dispersed in water, whereas the suspension in PGMEA tends to be quenched owing to the formation of GQD aggregates (i.e., the π‒π stacking interaction (ACQ mechanism)) [42]. Considering the GQD-100k suspension, the PEG framework tends to partially dissolve in high-polarity water molecules, causing the loss of GQDs from the cluster. Such a partial dissolution imparts a heterogeneous dispersion which disturbs the RIE structure and causes an unstable and weak PL emission under blue-light illumination. This undesired performance can be improved when the GQD-100k suspension is placed in low-polarity solvents, as shown in Figure 8b. The GQD-100k suspension in PGMEA maintains a well-developed framework in the solid state, where the rotational motion around the single bonds along with the partial energy relaxation (i.e., rotational and vibrational) promote the non-radiative decay pathways in the HOMO‒LUMO multiple band structures [1]. Within the PEG-coated GQD cluster, an aggregated solid state can be confined by the PEG nanostructure where intense intermolecular interactions occur between the GQDs and the PEG framework inside the aggregate that ultimately populates the radiative decay pathways, leading to the AIE fluorescence.

## 4. Conclusions

In this work, we have assessed the critical role of the solvent on the PL emission from PEG-coated GQD clusters under blue light and UV illumination. The one-pot hydrothermal method enabled the creation of PEG-coated GQD clusters using o-PD as the carbon and nitrogen sources in the presence of PEG with different molecular weights. Through analyzing various solvents, the AIE/ACQ behavior was determined for a series of as-prepared GQD structures. The PEG-coated GQD particles were suspended in ethanol, water, and PGMEA, enabling PL emission within the range of 530–550 nm. The pristine GQD suspension was quenched (i.e., QY: ~1%) when dispersed in PGMEA, resembling the ACQ mechanism. In contrast, the quantum yield was significantly intensified upon implementing the PEG nanocoating since the PEG layer promotes the AIE fluorescence process. Therefore, the one-pot hydrothermal route introduced in this work provides the initial framework for designing efficient ACQ/AIE cycles through modulating the interaction between the solvent’s molecular structure and GQD clusters. The findings of this study can be employed to synthesize finely tuned PEG-coated GQD clusters with numerous applications in optical, sensing, energy conversion/storage, and biological devices.

## Figures and Tables

**Figure 1 nanomaterials-11-01383-f001:**
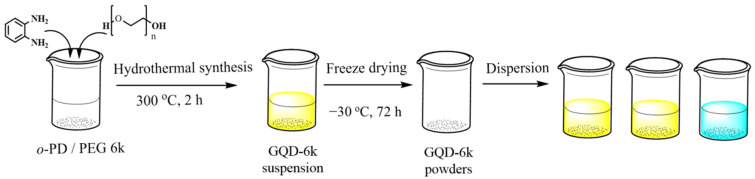
Schematic of PEG-coated GQD clusters (GQD-6k) through one-pot hydrothermal route.

**Figure 2 nanomaterials-11-01383-f002:**
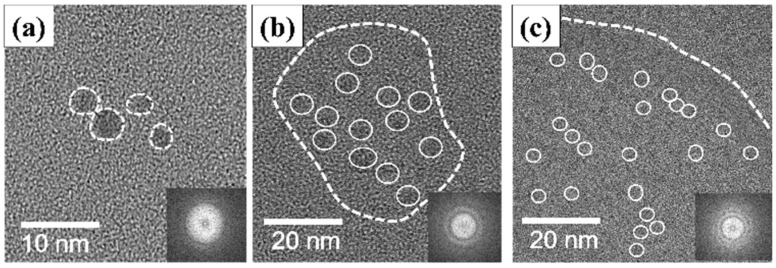
HR-TEM micrographs of (**a**) GQD-P, (**b**) GQD-6k, and (**c**) GQD-100k samples, where the inset shows their corresponding SAD patterns.

**Figure 3 nanomaterials-11-01383-f003:**
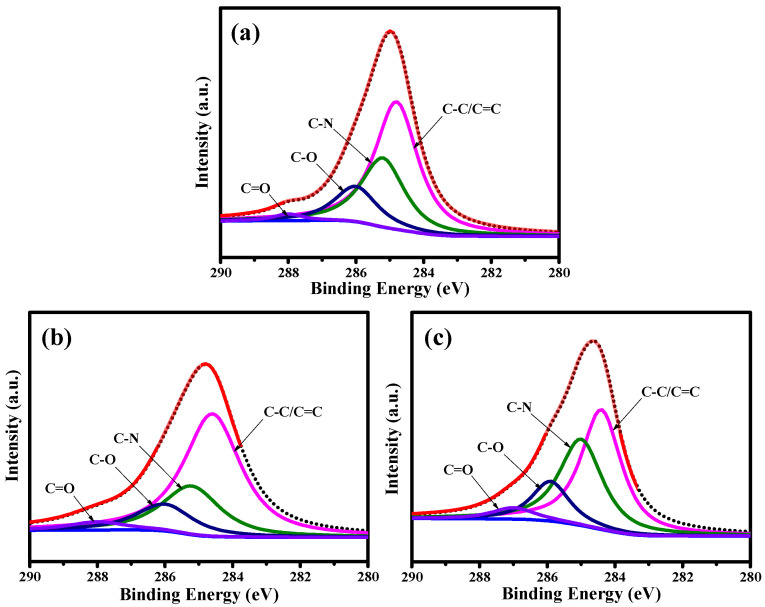
XPS C1s spectra of (**a**) GQD-P, (**b**) GQD-6k, and (**c**) GQD-100k samples, decomposed by a multiple Gaussian function.

**Figure 4 nanomaterials-11-01383-f004:**
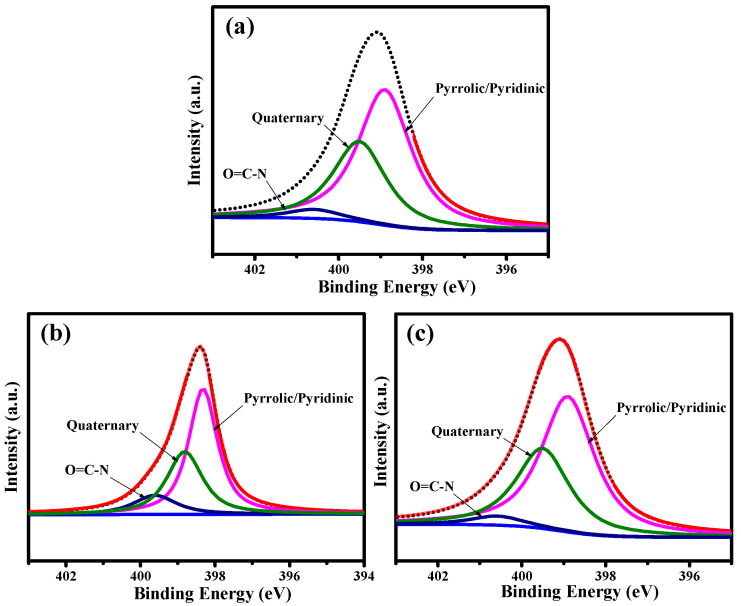
XPS N1s spectra of (**a**) GQD-P, (**b**) GQD-6k, and (**c**) GQD-100k samples, decomposed by a multiple Gaussian function.

**Figure 5 nanomaterials-11-01383-f005:**
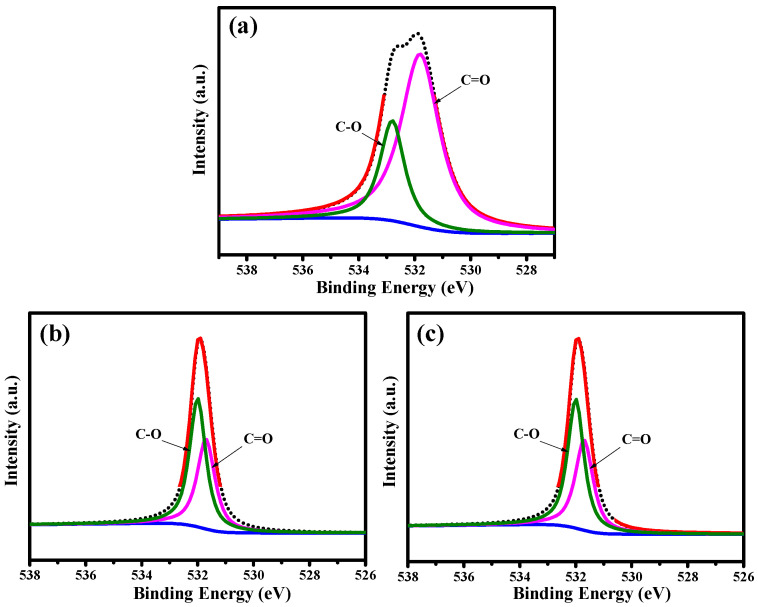
XPS O1s spectra of (**a**) GQD-P, (**b**) GQD-6k, and (**c**) GQD-100k samples, decomposed by a multiple Gaussian function.

**Figure 6 nanomaterials-11-01383-f006:**
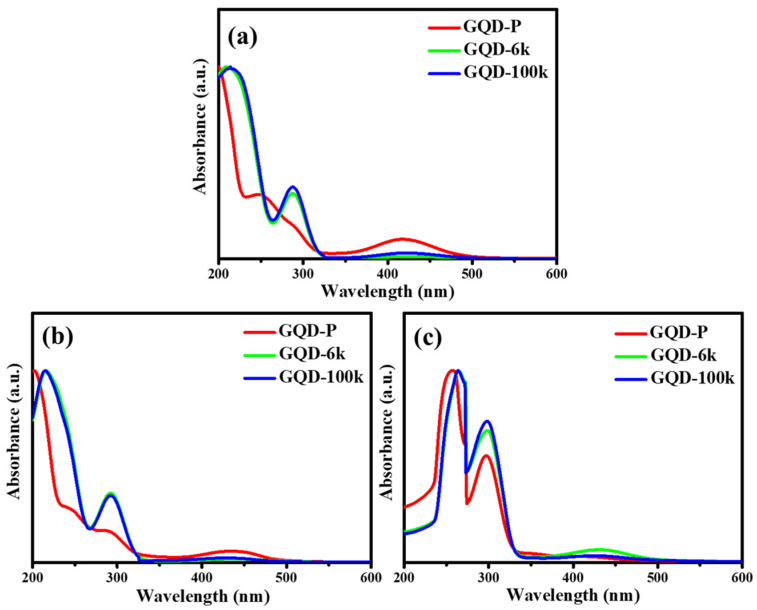
UV-vis absorbance spectra of pristine GQD and PEG-coated GQD cluster suspensions in (**a**) water, (**b**) ethanol, and (**c**) PGMEA.

**Figure 7 nanomaterials-11-01383-f007:**
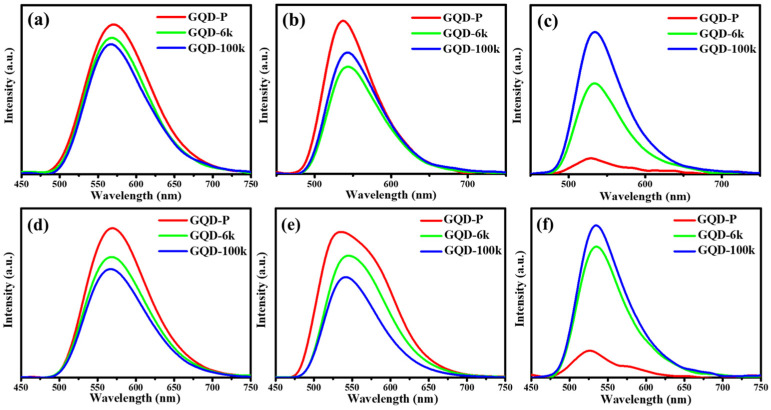
PL emission spectra of pristine GQD and PEG-coated GQD cluster suspensions in (**a**) water, (**b**) ethanol, and (**c**) PGMEA under an excitation of 360 nm. PL emission spectra of pristine GQD and PEG-coated GQD cluster suspensions in (**d**) water, (**e**) ethanol, and (**f**) PGMEA under an excitation of 450 nm.

**Figure 8 nanomaterials-11-01383-f008:**
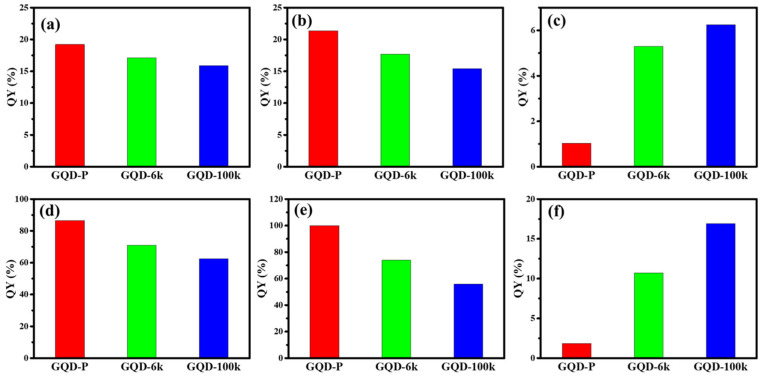
The QY values of pristine GQD and PEG-coated GQD cluster suspensions in (**a**) water, (**b**) ethanol, and (**c**) PGMEA under an excitation of 360 nm. The QY values of pristine GQD and PEG-coated GQD cluster suspensions in (**d**) water, (**e**) ethanol, and (**f**) PGMEA under an excitation of 450 nm.

**Figure 9 nanomaterials-11-01383-f009:**
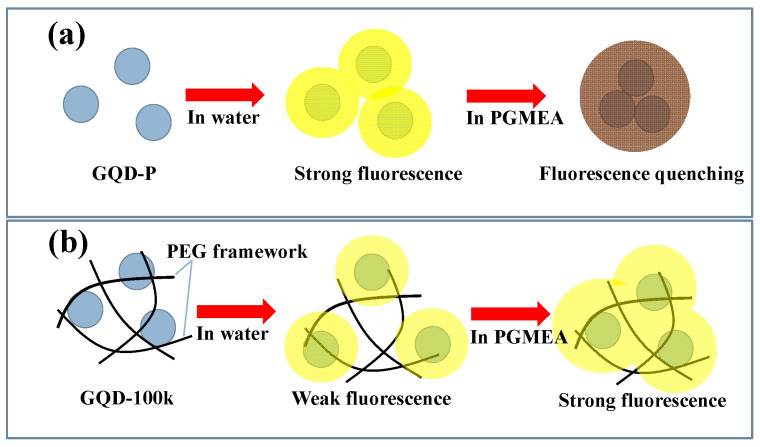
Schematic diagram for the PL emission from (**a**) GQD-P and (**b**) GQD-100k suspensions in different solvents under blue-light illumination.

## Data Availability

Data is contained within the article or Appendix A.

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
