# Peer review of "Effect of Solvent on Fluorescence Emission from Polyethylene Glycol-Coated Graphene Quantum Dots under Blue Light Illumination"

_nanomaterials, 2021, doi:10.3390/nano11061383_

Round 1

Reviewer 1 Report

This manuscript reports on the effect of different solvents on the fluorescence emitted by Graphene quantum dots coated with PEG for excitation with blue and UV light. The manuscript is similar to a previous manuscript by the same group on PEG-coated CQD clusters, although novelty is clearly stated. There are some questions and points that need to be clarified/improved:

  1. In the abstract and in the “materials and methods” section it is mentioned that three different solvents (PGMEA, ethanol and water) were used for dispersing the PEG-coated GQDs while in lines 80-82 it is mentioned that the effect of molecular weight of PEG on the PL emission from the PEG-coated GQD clusters was explored in water, ethanol, PGMEA and n-hexane. Finally it seems that measurements were done only in three solvents. Please check and correct.
  2. It would be appropriate to include the “materials and methods” section before the “results and discussion” section taking into account that in the “materials and methods” section the labels used to identify each sample are defined
  3. Figure 1d does not exist (line 102)
  4. In lines 135-136 it is stated that from figure 2 it is very clear that the oxygen functionalities decrease upon introducing PEG into the GQD clusters but from the graphs it is not a clear conclusion. Can this be quantified? In fact, immediately after it is mentioned that “the PEG framework provides a large amount of H−(O−CH2−CH2)n−OH chains, covering the GQD solid state”, and then, does it make sense that the amount of oxygen functional groups decrease?
  5. Regarding XPS O1s spectra: it seems that C-O contribution increases in comparison to C=O contribution upon introduction of PEG into the clusters. Please explain.
  6. Concerning FTIR spectra it is mentioned that “band ca. 3250-3400 cm-1 is due to the adsorption of water molecules or N‒H group on the surface of GQD clusters” but PEG also contributes with OH groups. Also more description should be given in the explanation of these results and the comparison between pristine GQD and PEG-coated GQD.

Author Response

Response to Reviewer #1:

         We would like to express our gratitude for the reviewer’s effort to improve the quality of this manuscript. What changes of the manuscript have been made are shown in the following responses to the specific comments. The corrections in the revision have been indicated in blue.

  1. Page 2. The reviewer’s inspection is right. In the present manuscript, the PL measurements were carried out in three solvents, namely, water, ethanol, and PGMEA. We have corrected this typo in the revision.

  1. Page 2-3. The reviewer’s concern is appropriate. For good readability, we have moved the “Materials and Methods” section before “Results and Discussion”, according to the reviewer’s suggestion. We have reflected it in the revised manuscript.

  1. Page 4. The reviewer is right. The typo has been corrected in the revised manuscript.

  1. Page 4 and 5. The reviewer’s concern is appropriate. To avoid reader’s confusion, one brief description regarding the XPS measurements was added into the revised manuscript. We feel that the adjustable oxidation/amidation level is mainly due to the formation of the PEG skin layer that covers the GQD particles and constructs a stereo framework of H−(O−CH2−CH2)n−OH chains. The decreased O/C ratio after the PEG coating is presumably due to the polymer chain containing a large number of alkyl groups that tend to totally cover O-rich GQD surface (i.e., GQD-P (O/C ratio: 53.6 at.%)). We also observe that the O/C ratio is a decreasing function of PEG content. In addition, Due to the PEG skin layer, the C‒O/C=O ratio shows an increasing trend with the PEG content, i.e., GQD-P (0.59) < GQD-6k (1.63) < GQD-100k (1.86). We have reflected it in the revised manuscript.

  1. Page 7. The reviewer’s suggestion was adopted. We have briefly described the FT-IR spectra in the revised manuscript, as follows: “…a single band at ca. 3250‒3400 cm-1 is observed due to the adsorption of water molecules or N‒H group on the surface of GQD clusters [34]. Due to the hydrophilic PEG skin layer, the transmittance band becomes more evident on the PEG-coated GQD clusters compared to the pristine GQD sample, indicating the presence of water-adsorbed layer onto the PEG-coated GQD clusters.”. We have reflected it in the revision.

Reviewer 2 Report

I have read the manuscript “Effect of Solvent on Fluorescence Emission from Polyethylene Glycol-Coated Graphene Quantum Dots under Blue Light Illumination” by Po-Chih Yanga et al. (MS # nanomaterials-1211615) submitted for the publication in Nanomaterials.

The authors report the results on the preparation and characterization of PEG-coated GQD clusters, with photoluminescence under the blue light. They found that the QY of such suspensions was dependent on the solvent type.

The paper is well written and interesting as the increase of PL emission is a challenge for the next-generation optical devices. Consequently, I recommend the publication of the manuscript in Nanomaterials as it is.

Author Response

Response to Reviewer #2:

         We would like to express our gratitude for the reviewer’s effort to improve the quality of this manuscript. What changes of the manuscript have been made are shown in the following responses to the specific comments. The corrections in the revision have been indicated in blue.

We deeply appreciate the reviewer’s encouragement. The future work regarding the optical detection of metal ions and bio-imaging applications is on-going. 

Reviewer 3 Report

  1. The Introduction must be more clearly explained and analyzed and new references should be added, particularly some very recent ones.
  2. I am not so sure whether the new items of the paper are would be well understood by the readers. Please try to be more specific on this issue.
  3. The third section of the paper is rather small. It should be expanded somehow in order to provide additional information.
  4. Are the proposed methods only applicable to this particular type of graphene?
  5. The quality of all pictures should be enhanced. Perhaps it is my pdf file. If not, please try to improve it.

Author Response

Response to Reviewer #3:

         We would like to express our gratitude for the reviewer’s effort to improve the quality of this manuscript. What changes of the manuscript have been made are shown in the following responses to the specific comments. The corrections in the revision have been indicated in blue.

  1. Page 2. The reviewer’s concern is appropriate. The Introduction has been moderately modified in the revision. An extensive literature review regarding the fluorescence from GQDs under blue-light illumination has been inserted into the Introduction section of the revised manuscript. Five new references have been added into the revised manuscript, and thus the reference list was reordered in the revision.

  1. The reviewer’s suggestion was adopted. For good readability, we went through the whole manuscript and made some modifications on some sentences.

  1. Page 3. We adopted the reviewer’s suggestion. For good readability, we have moved the “Materials and Methods” section before “Results and Discussion”, according to the reviewer’s suggestion. One brief description was added into the revised manuscript to enrich this section of the revised manuscript. We have reflected it in the revised manuscript.

  1. Page 2. The reviewer’s concern is creative. So far, we have successfully prepared highly luminescent GQDs coated with PEG skin layer using modified pyrene precursors through the solvothermal synthesis method. For comparison, Ref. [19] (concerning the solvothermal synthesis of functionalized GQDs from modified pyrene precursor) has been cited in the revised manuscript. We have reflected it in the revision.

  1. The reviewer’s inspection is right. For clarification, we have carefully pasted high-resolution images in the revised manuscript.

Round 2

Reviewer 1 Report

All the questions have been answered and there are no additional comments

Reviewer 3 Report

In my opinion, the authors have improved their paper and it can now be deemed mature for publication.